# A 3D-CT Analysis of Femoral Symmetry—Surgical Implications

**DOI:** 10.3390/jcm9113546

**Published:** 2020-11-03

**Authors:** Joan Ferràs-Tarragó, Vicente Sanchis-Alfonso, Cristina Ramírez-Fuentes, Alejandro Roselló-Añón, Francisco Baixauli-García

**Affiliations:** 1Department of Orthopaedic Surgery, Hospital Universitario y Politécnico La Fe, Avinguda de Fernando Abril Martorell, 106, 46026 València, Spain; baixauli_fragar@gva.es; 2Department of Orthopaedic Surgery, Hospital Arnau de Vilanova, Carrer de Sant Clement, 12, 46015 València, Spain; vicente.sanchis.alfonso@gmail.com (V.S.-A.); alexrosello82@gmail.com (A.R.-A.); 3Department of Radiology, Hospital Universitario y Politécnico La Fe, Avinguda de Fernando Abril Martorell, 106, 46026 València, Spain; crisramirezfuentes@gmail.com

**Keywords:** femoral torsion, femoral anteversion, femoral maltorsion, CT, 3D-CT, osteotomy, patellofemoral

## Abstract

Background: Mirroring the image of the affected side is a widely used technique for surgical planning in orthopedic surgery, especially for fractures and custom-made prostheses. Our objective is to evaluate the three-dimensional symmetry of the femurs using finite element analysis and manual alignment. Methods: Using the computed tomography of 15 patients without lower limb pathology, 30 3D biomodels of their femurs were obtained. The error obtained through image manipulation was calculated and broken down into a rendering error and a manual overlay error. The Hausdorff–Besicovitch method was applied to obtain the total asymmetry. The manipulation error was theb subtracted from it to obtain the intrapersonal asymmetry. Results: The mean intrapersonal asymmetry was 0.93 mm. It was obtained by subtracting the error derived from rendering and alignment of 0.59 mm (SD 0.17 mm) from the overall mean error of 1.52 mm (SD 1.45). Conclusions: Intrapersonal femoral asymmetry is low enough to use the mirror image of the healthy side as a reference for three-dimensional surgical planning. This type of planning is especially useful in deformity surgery when the objective of the surgery is not to restore only one specific parameter but to obtain a general functional morphology when a healthy contralateral femur is available.

## 1. Introduction

### 1.1. Background

Preoperative planning in orthopedic surgery and trauma reached a new height with the revolution that 3D imaging brought on. New software programs allow for the management of 3D images in an economical and simple way. It has made for the routine use of this technology in many orthopedic surgery and trauma departments worldwide. Custom-made prosthetic implants [1,2,3], pre-modeling of osteosynthesis plates [4,5,6], and preoperative planning [7,8] are some of the uses of 3D technology in orthopedic surgery and trauma. In practically all of them, the use of a mirror image of the healthy side is taken to establish the working normal for each patient [9,10]. In other words, it is the morphological aim that we seek to achieve with corrective surgery.

Although some authors have observed that both femurs of an adult are symmetrical [11], others have found a clear asymmetry between the two [12,13]. This has clinical implications in regards to the use of the contralateral femur as a reference standard when going about surgical planning. This asymmetry will be at fault for the error that we generate when using the mirror image of the healthy side as a reference in three-dimensional surgical planning. The asymmetry between the two femurs may be a consequence of an underlying unilateral pathology or the method used to assess symmetry. Our working hypothesis is that if we ignore these biases by including only those patients with no previous history of documented pathology in the lower limbs in our study and using a proven method of evaluating the similarities of volumetric structures, both femurs of the same person will be symmetrical. For this study, we used the Hausdorff-Besicovitch method [14], which is widely used in engineering to evaluate the morphological discrepancy between two bodies in a three-dimensional workspace. The objective of this study is to evaluate the symmetry of both femurs in adults to determine the error made when using the healthy side as a reference in the planning of derotational femoral osteotomies.

### 1.2. Rationale

Intrapersonal 3D symmetry in the femur has been widely analyzed using automatic alignment and rendering tools. These tools align two femurs as perfectly as they can, but we cannot know the alignment points used during the comparison. Therefore, it is not useful to analyze the core of rotation in femoral maltorsion.

This is the first study to analyze the intrapersonal symmetry of the femur using a manual alignment tool based on the anatomic definition of femoral anteversion. For this reason, when both femurs are properly aligned manually, we can be sure that the differences found are due to the intrapersonal asymmetry and are not modified by the automatic alignment tool. This explains the discrepancies between the degrees of asymmetry found in other studies, finding a solution and an explanation for all of them.

## 2. Experimental Section

To quantify the asymmetry between a femur and the mirror image of the contralateral side in millimeters, the process started with angio-CTs performed in diabetic patients for the study of peripheral vascular disease and excluded those with a previous history of documented osteoarticular pathology in the lower limbs.

The population included in the study was considered normal due to the absence of clinical findings in the retrospective review of their medical histories and the normal values of the cervico-diaphyseal angle and femoral rotation angle. To evaluate the intrapersonal cervico-diaphyseal angle differences. a Wilcoxon analysis was applied and their differences were assumed statistically significant when the *p*-value was higher than 0.05. To evaluate the intrapersonal femoral rotation, Murphy’s method was used [15] and the Wilcoxon analysis was selected to analyze the intrapersonal differences between both femurs, considering a *p*-value higher than 0.05 to be statistically significant. The mean cervico-diaphyseal angle on the right side was 126.26° SD 6.21° IC95% (122.95–129.57° *p*-value ˂ 0.01) and on the left side, the mean cervico-diaphyseal angle was 125.73° SD 5.98° IC95% (122.54–128.92° *p*-value ˂ 0.01). The mean intrapersonal differences on the femoral cervico-diaphyseal angle was 2.4° SD 1.88° IC 95% (1.35–3.44° *p*-value ˂ 0.01), and these differences were not statistically significant (*p*-value > 0.05). The mean femoral torsion on the right side was 19.27° SD 7.41° IC95% (14.5–23.98° *p*-value ˂ 0.01) and on the left side, the mean femoral torsion angle was 16.27° SD 7.18° IC95% (11.7–20.83° *p*-value ˂ 0.01) The mean absolute intrapersonal difference on femoral rotation was 4.45° SD 3.17° IC 95% (2.32–6.58° *p*-value ˂ 0.01), and these differences were not statistically significant (*p*-value > 0.05). According to the data, the patients were considered normal patients.

The 3D biomodel of both femurs was rendered to obtain the volumetric values with the Threshold Effect tool based on the Hounsfield units and using the same value for both femurs (3D Slicer^®^ Harvard Medical School, Massachusetts MA, USA). Subsequently, elements other than the femur (pelvis, arteries, patella, and tibia) were removed using MeshMixer (Autodesk Inc^®^, San Rafael, California CA, USA) without making additional solidification corrections or defect closure to avoid the errors associated with post-rendering modification.

Next, the mirror image of the left side was made using 3D Builder (Microsoft Corporation^®^, Washington WA, USA) (Figure 1). Both femurs were aligned, taking the horizontal plane as the neutral reference plane and marking the contact points of the femoral condyles and the trochanteric mass with the horizontal plane. These points are the one of the most popular references used by anatomists while studying femoral anteversion in cadaveric specimens and they can be easily reproduced in 3D programs. This was done to establish easily reproducible reference points for manual alignment of both femurs, saving the images in the stereolithography (.stl) format (Figure 1, Appendix A).

Finally, the images of the right femur and the specular image of the left femur were imported into the MeshLab program (Visual Computing Lab^®^, Institute of the National Research Council of Italy, Pisa, Italy) and manual alignment of both femurs was performed using the marked contact points with the horizontal plane of the previous step [17]. Once the femurs were aligned, the Hausdorff-Besicovitch method of analysis was applied to calculate the mean of the differences, the maximum difference, and its variability between the two femurs [14].

The sample size was established in 30 biomodels based on a β error of 0.2, a significance value of *p* < 0.05, a standard deviation of 7 mm, and an accuracy of 5 mm [18].

We must bear in mind that when the method Hausdorff-Besicovitch is applied in the described process, there are three steps in which an error with the process itself can be generated. Erroneous rendering can come about that generates a discrepancy between both femurs without actually being a discrepancy. An error can also be generated when the two femurs are manually aligned. For this reason, when the Hausdorff-Besicovitch method was applied to the comparison, the discrepancy between structures that we objectified was due to the sum of all the errors included in the process, the rendering error, the alignment error, and the error due to the patient’s own asymmetry. To obtain the error simply dur to asymmetry, the alignment error and the rendering error was added together and then subtracted from the total error that we got during the comparison.

To quantify the alignment error, the 15 right femurs were rendered once, and the biomodel was obtained by following the steps previously described (Figure 1, Appendix A). Subsequently, the right femur was aligned over the same right femur, and the discrepancy between the two femurs was calculated using the Hausdorff-Besicovitch method. If the alignment was perfect, no difference would be seen since it was the same patient and there was no anatomical variability as the alignment was made with a biomodel of the very same femur. Thus, there were no differences in rendering since it was the same. In this way, the average error attributable to the manual alignment process was acquired. To calculate the rendering error, the same CT was rendered in duplicate to obtain two biomodels of the same femur. This was done on the 15 right femurs to obtain 15 pairs of femurs. Once the two biomodels of the same femur were obtained, they were aligned, and their discrepancies were analyzed with the Hausdorff-Besicovitch method. The discrepancy would be the result of the sum of the rendering and alignment errors, since the asymmetry error did not exist, as it was the same femur from the same patient. Because the value of the mean alignment error was obtained in the previous step, the expected average error attributable to the rendering was given when we subtracted the average error obtained in this step from the average alignment error. This process was carried out on 15 patients, and the mean rendering error was taken as the mean error of the 15 measurements. Once the isolated values of these errors were obtained, they were subtracted from the mean error obtained in the comparison between the right femur and the mirror image of the left side to isolate the difference due to asymmetry.

The Hausdorff-Besicovitch method evaluates asymmetry in millimeters. To convert the millimeters of difference to degrees, the length of the femoral neck was calculated and the conversion to degrees of the error produced was obtained (Figure 2)

The study was approved by the ethics committee of the institution (2020-277-1).

## 3. Results

The mean alignment error was 0.32 mm, the SD was 0.22 (95% CI 0.19–0.44 mm, *p*-value ˂ 0.01), and the mean of the maximum alignment error was 1.34 mm SD 1.02 mm (95% CI 0.77–1.91 mm *p*-value ˂ 0.01) (Figure 3 and Figure 4).

The mean rendering error was 0.59 mm, the SD was 0.17 mm (IC95% 0.37–0.8 mm *p*-value ˂ 0.01), and the mean maximum error was 6.63 mm SD 2.12 mm (95% CI 4–9.27 mm *p*-value ˂ 0.01) (Figure 3 and Figure 4).

The mean total error between a femur and the mirror image of the contralateral side after the process was 1.52 mm SD 1.45 mm (95% CI 0.71–2.33 mm *p*-value ˂ 0.01), and the mean maximum error was 8.5 mm SD 2.89 mm (95% CI 6.94–10.15 mm *p*-value ˂ 0.01) (Figure 4). The mean error due to asymmetry was calculated at 0.93 mm, subtracting the rendering error and alignment error from the total error (Figure 4 and Figure 5).

The mean of the intrapersonal differences of the femoral version evaluated with the described method were 1.13° SD 1.06°, while the interpersonal variability of the femoral version was 5.2° SD 3.49, this difference being statistically significant (*p*-value ˂ 0.01). This implies that the error made when restoring the femoral version of a subject, using the population mean as a reference, was greater than the error made using the described method (Figure 5).

The described method evaluated an average of 891.730 anatomical references in each of the comparisons, the minimum value of the references being 260.685 points.

## 4. Discussion

The main finding of this study was that there is a high degree of similarity between the right and left femur in a healthy adult. The mean overall asymmetry between the two femurs of the same patient is, on average, approximately 1 mm, with the mean maximum asymmetry being 8 mm. Translated into angles, we would restore the normal anatomy for that patient with a total mean margin of error of 1° of femoral anteversion and an approximate maximum mean error of 5° if we were to use the healthy side as a reference for correction in a unilateral derotational femoral osteotomy. This confirms the initial hypothesis of the work and contributes new ways of using the three-dimensional image for preoperative planning of femoral torsional deformities.

Virtual three-dimensional osteotomies on the pathological side can be planned up to the point of getting its three-dimensional structure to be identical to the mirror image of the healthy side [19,20,21]. If we were able to sculpt an anteverse femur until it was identical to the mirror image of its healthy contralateral side, we would know that we have obtained a femur very similar to what the patient should have had from this data. There would be an average error of 1 mm in any of the points of its entire structure. Until now, preoperative planning has been based on the restitution of some of the morphological parameters, for example femoral anteversion of the femur. With this new system, nearly 900,000 anatomical points were computed in each planning process, which increased the accuracy of the correction and allowed for a restitution ad integrum of the morphology that the patient should have had without a pathology.

Currently, there is controversy surrounding symmetry between the two sides. Authors, such as Eckhoff et al. [13] and Dimitrou et al. [12], defend the existence of an approximate difference of 5° in femoral anteversion using three-dimensional technology to assess the angle of femoral anteversion. On the other hand, authors, such as Bakhsayesh et al. [11], use 3D global morphological assessment systems and argue that this symmetry is not so great. The reason for this discrepancy is that global volumetric assessment tools, like the one used in this study, assess a large number of asymmetry points, including the entire femur. Therefore, in cases in which the majority of the asymmetry is concentrated on a specific point of the femur (in the neck, for example), the total asymmetry is reduced by including points with less asymmetry, such as the diaphysis or the femoral condyles in the comparison.

In line with previous studies, most of the femoral rotational deformity was located at the level of the femoral neck. Our data allowed us to explain the discrepancy between the published asymmetry studies. By aligning both femurs on the horizontal plane, we obtained a mean maximum discrepancy of approximately 5° that was related to the proximal femoral asymmetry described by other authors, as well as a total mean difference of approximately 1° by including the diaphysis and distal femur in the evaluation.

Therefore, the use of specular imaging of the healthy contralateral side with manual alignment is a useful tool for analyzing the origin of deformity and osteotomy planning (Figure 6). Combining this tool with the new three-dimensional tools for measuring the femoral anteversion angle, the planning of both the site and the magnitude of the osteotomy can be optimized.

### Limitations

The main limitation of this method is the low prevalence of unilateral femoral torsional deformities. On the other hand, it is not uncommon to see patients with an abnormally high bilateral femoral anteversion and unilateral symptomology. In these cases, due to ignorance of the etiopathogenesis of pain, one cannot be sure of the clinical resolution of pain in the symptomatic limb by using the asymptomatic side as a morphological reference. Therefore, we usually prefer to return the anteversion values to mean population values without being sure that this will produce a clinical resolution. At the method level, the error produced during obtaining the biomodel could not be completely, even though the isolated asymmetry was less than 1 mm. It could be reduced by obtaining higher resolution tomography images. However, even considering the error derived from the image manipulation process, the total error was less than the error that was produced when considering the population mean for the restitution of the femoral version angle. Consequently, it is not clinically necessary to obtain better CT resolution than that which is normally used.

Another limitation of the study is that the CTs were analyzed retrospectively from a group of the population that carried out a CT due to a different situation, and the clinical information was obtained retrospectively. Nevertheless, we did not find differences in cervico-diaphyseal or femoral torsion angles between the right and left sides; therefore, we must assume that the asymmetry found in our data was the asymmetry that we could find in the normal population.

Finally, we must highlight that manual alignment tools have an intrinsic intra and interobserver variability. Even if comparing manual and automatic alignment tools is not the aim of this work, it must be considered when analyzing the results, taking into account that the intra or interobserver agreement could modify the intrapersonal asymmetry according to the confidence interval expressed in the results. To evaluate the interobserver agreement, more studies that analyze this point specifically are needed.

## 5. Conclusions

The Hausdorff-Besicovitch method, widely used in engineering to assess the morphological discrepancy between two bodies in a three-dimensional space, is an accessible preoperative planning methodology in surgery for femoral torsional deformities. It makes an anatomical restitution practically identical to what the patient should have had, with a medically insignificant mean error, in the absence of a pathology. A symmetry between both femurs has been seen that supports the use of the contralateral femur as a model for preoperative 3D planning.

## Figures and Tables

**Figure 1 jcm-09-03546-f001:**
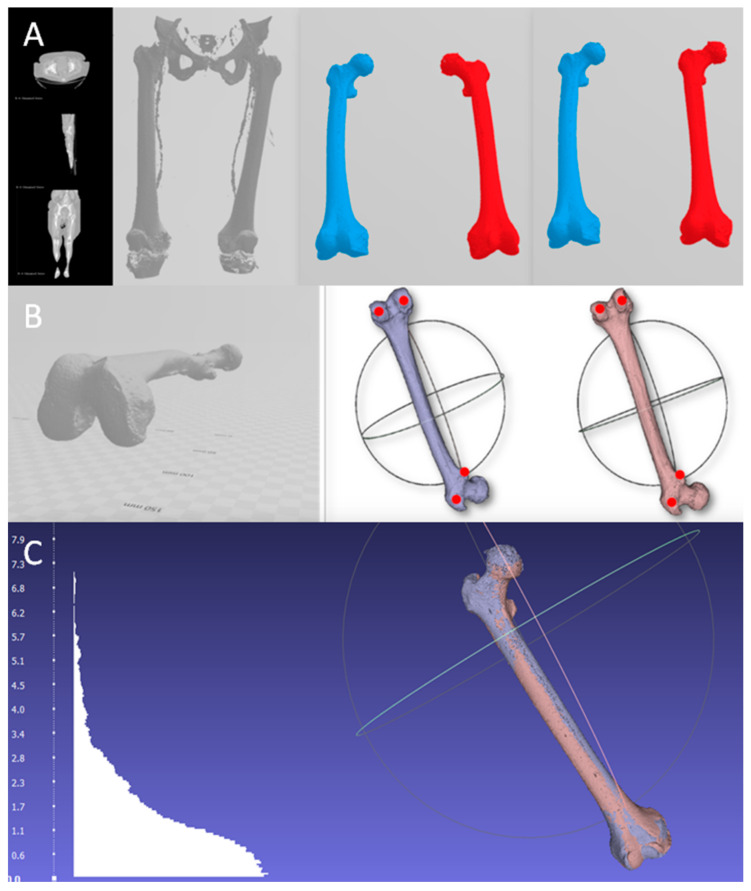
Steps to obtain the model and its alignment. (**A**) The 3D biomodel is obtained from the tomography, from which the right (blue) and left (red) femur are isolated to subsequently perform the mirror image of the left side. (**B**) Alignment with the horizontal of the femur and marking the reference points for subsequent alignment. To establish a common horizontal plane to all the points that will be used during the manual alignment, the table top method was used [16]. (**C**) Result after superimposing both femurs and the histogram with the differences of the anatomical points compared.

**Figure 2 jcm-09-03546-f002:**
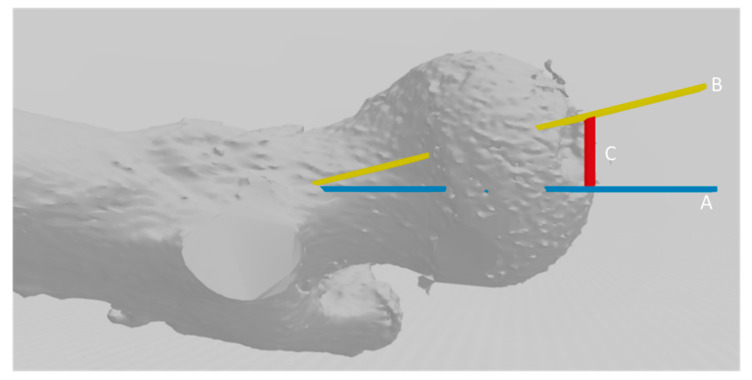
Extrapolation of the error in millimeters calculated with the Hausdorff-Besicovitch method in degrees of femoral anteversion using Murphy’s method [15]. Line C, in red, represents the magnitude of the error. Line B represents the femoral neck, whose value can be calculated. After calculating the length of the femoral neck, it acts as a hypotenuse of a right-angled triangle whose right angle is represented between lines C and A. Therefore, the sine of angle B–A is obtained by dividing the value of length C/length B, whose data are known. Therefore, it converts the error in millimeters into angles of error to be able to test the hypothesis with the degrees of errors produced, taking the population mean as a reference.

**Figure 3 jcm-09-03546-f003:**
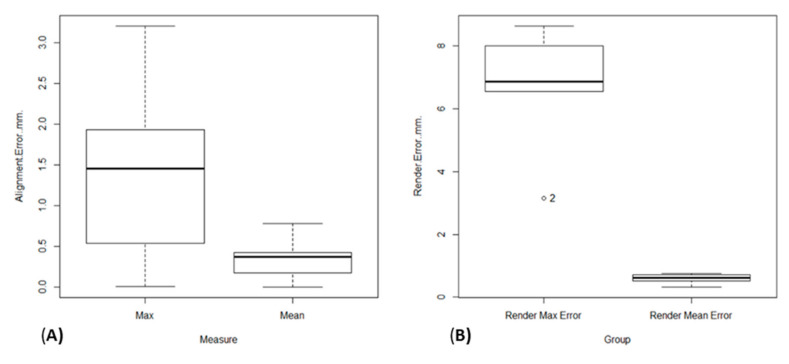
Average and maximum error in the alignment step (**A**); average and maximum error in millimeters in the rendering step (**B**).

**Figure 4 jcm-09-03546-f004:**
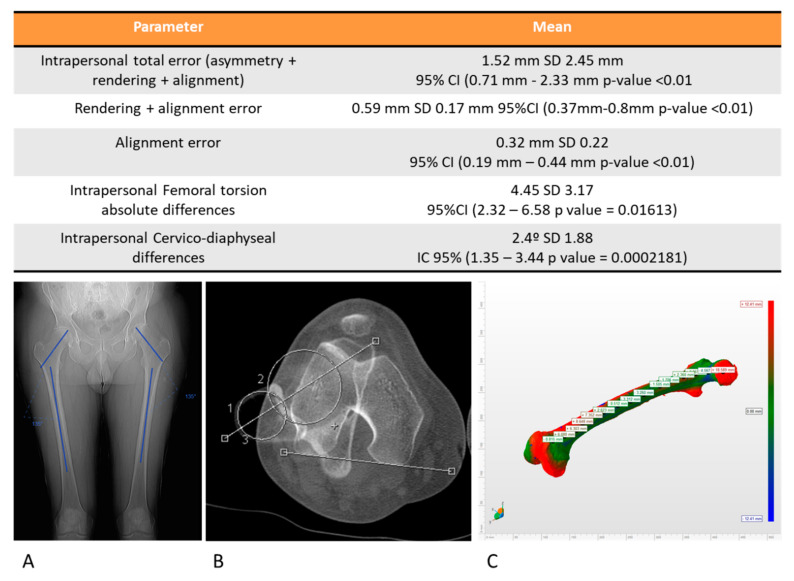
Table of the results. (**A**) Measurement of the intrapersonal cervico-diaphyseal angle asymmetry. (**B**) Measurement of the femoral torsion using Murphy’s method. (**C**) Representation of the measurement of the intrapersonal 3D asymmetry.

**Figure 5 jcm-09-03546-f005:**
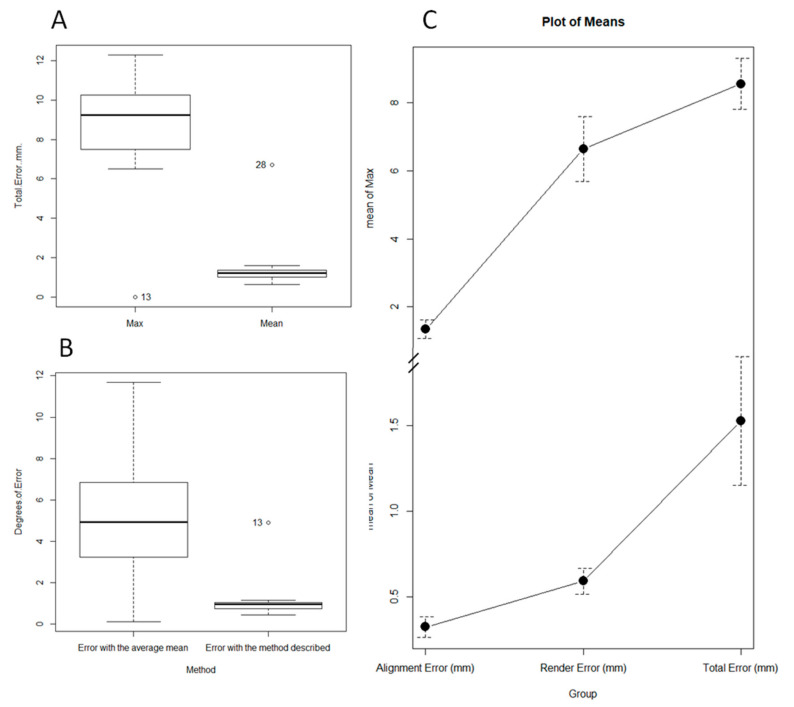
(**A**): Maximum mean error and total mean error between a femur and the mirror image of the contralateral side. These errors include the skewness error, the rendering error, and the alignment error, intrinsic to biomodel processing. (**B**) Interpersonal variability (represented as the mean of the differences in the femoral version of the patients, with respect to the series mean) and intrapersonal variability (represented as the mean of the differences in the femoral version of the patients, with respect to their contralateral side with the described method). (**C**) Representation of the contribution to the total error of the alignment error and the rendering error.

**Figure 6 jcm-09-03546-f006:**
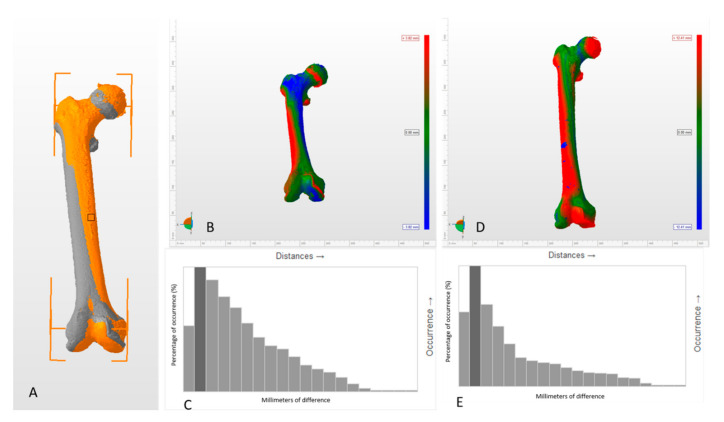
Changes in asymmetry pattern between manual and automatic alignment methods (Autodesk Netfabb 2019, Autodesk, California CA, USA). (**A**) General view of both femurs manually aligned. The right femur is in grey and the left femur is in orange, both overlapped. (**B**) Asymmetry pattern between two femurs manually aligned. The differences between both femurs are represented in a color code that is referenced in the lateral bar. The intensity of the blue and red colors represents the magnitude of the positive and negative differences between both femurs, respectively. The green color represents the absence of differences. (**C**) Histogram of the behavior of the differences between two femurs manually aligned. (**D**) Asymmetry pattern between two femurs automatically aligned. The main limitation of the automatic alignment method is that we cannot know in which position the software has aligned both femurs, and, consequently, it is not useful to analyze the origin of the deformities. (**E**) Histogram of the behavior of the differences between two femurs automatically aligned. Histograms are shown just as a visual explanation of the discrepancy on the distribution of the differences between the two alignment modes, but it is not the aim of this work to evaluate which of the method is the best, since, depending on the purpose, the method of choice can vary. However, it is a simple way to show how the alignment method can modify the interpretation of the results.

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
