# Peer review of "A 3D-CT Analysis of Femoral Symmetry—Surgical Implications"

_jcm, 2020, doi:10.3390/jcm9113546_

Round 1

Reviewer 1 Report

Very good study and also very worthwhile. We often work with the assumption that the left and right sides are fairly symmetrical when using the mirrored side as a reference to the side to be repaired. It is nice to see a study that provides objective numbers rather than the subjective “yep, looks the same.”

The introduction is good and provides enough background. The methodologies are sound. The results and the conclusions drawn from those results are supported.

There is an occasional odd grammar error, a period (.) at times appears randomly. See first paragraph of 2. Experimental section “in the lower limbs. . The 3D…” However, the English is very good.

Nice work.

Author Response

Dear Reviewer

Thank you so much for your feedback on our work. We didn’t notice the additional stops in these phrases.

We hope that with this work and your recommendations we will help to the orthopedic surgeons to improve the way they treat their patients and to better diagnose this disease. Thank you so much

Joan Ferràs-Tarragó

Vicente Sanchis-Alfonso

Reviewer 2 Report

In this paper, the authors evaluate the intrapersonal femur symmetry of 15 subjects in the context of orthopaedic surgery planning, where the anatomy of the healthy side is usually used as a reference to restore the native anatomy of the diseased hip/femur. The authors focus on the specific case of derotational femoral osteotomies and in patients with no underlying unilateral pathology. The symmetry evaluation is performed doing a manual alignment of both femurs and computing the Hausdorff distance.

I consider that this paper needs to address some major issues in order to be ready for publication:

  • The authors claim that manual alignment of the femurs is better than automated methods. However, manual alignment is user dependent and subjective. In my experience, automated methods work better for femoral alignment and more importantly, they are based in a quantitative objective metric. If the authors quantitatively justified the use of manual alignment, they would need to include an inter and intra-user reliability analysis.
  • The authors use the Hausdorff-Besicovitch method to quantify the asymmetry, which is a good and standard method to compare two different volumes or surfaces. However, this is a global metric that can easily masquerade some local asymmetries and it does not have an anatomical/functional meaning. I think it would be important to incorporate anatomical femoral measures (such as version or neck-shaft angle) to perform a better comparison.
  • The study design including only patients that went through angio-CTs performed in diabetic patients without a previous history of documented osteoarticular pathology in the lower limbs is detrimental for this paper, which claim to focus on case of derotational femoral osteotomies. If this cannot be solved, a good discussion as how these results from this study sample are meaningful and important for planning derotational femoral osteotomies.

Author Response

Dear reviewer

First of all thank you so much for your feedback. With your excellent comments, we will improve it and will be easier to understand for the readers.

1-. We agree with you that the manual alignment tool has an intra and interobserver variability. We didn’t include a specific analysis of it because it isn’t the aim of the work to evaluate which alignment tool is better but to evaluate how could if affect to the asymmetry quantification. In the work, the manual error is considered as one of the errors included in the process so, in case the intra or interobserver manual error was different, it would affect the total error value, but in this work, we define the asymmetry error after subtraction of the manual error so, independently of the value of the manual alignment, the asymmetry error mustn’t be different.
We have included this statement in the discussion as a limitation to clarify it, thank you so much (277-280)

2-. Completely agree with you, we have included some information about the anteversion values and cervico-diaphyseal angle of the patient of the serie (72-87) and a new table to resume all the results (Figure 5). They are not quite different of the results published before in the literature, but to better understand the work, these results will help, thanks

3-. To evaluate the volumetric asymmetry, we needed 3D CT of the complete femur, what implies a high radiation dose. That’s the reason why we decided to do this work retrospectively. This is the first work that uses this methodology, and their results are promising to do prospective analysis using the same methodology, but if the method hadn’t been useful, we would have exposed to lot of people to an unnecessary CT. We agree that the population whom the CT are used must be clarified, and we have added a comment in the discussion (271-275), thank you so much.

Finally, I would like to thank you your time and effort helping us to improve our work. This work is a bit different in respect with the traditional point of view of orthopedic surgery measurements, due to its 3D comprehension of the bones. We understand that I can be difficult to understand but thanks to the new software and technology, the 3D analysis will become more popular in the next few years, and with our results, we can stablish which is the error that we commit when taking the mirror image of the healthy contralateral bone as reference, which is one of the most used references in orthopedic surgery and traumatology. With your help our work has improved significantly, thank you so much.

Joan Ferràs-Tarragó

Vicente Sanchis-Alfonso

Reviewer 3 Report

jcm-971980 A 3D-CT Analysis of Femoral Symmetricity – Surgical Implications

This is a scientific study evaluating side-to-side differences in femoral anatomy by computer analysis of segmented 3D models from CT of subjects without lower limb pathology. The study is interesting to read with credible results and provides further insight into the use of the contralateral femur for templating in various forms of hip surgery.

General: The language is mostly good, however with some colloquialisms, superflous spaces and periods. The noun symmetricity should be changed to the simpler symmetry in the title and manuscript.

Introduction

Although not clearly stated, the only symmetry reported is that of femoral neck version. Other important measures of symmetry in total hip arthroplasty such as acetabular offset and true femoral offset are not mentioned (see e.g. Geijer et al. 2020 Pre- and postoperative offset and femoral neck version measurements and validation using 3D computed tomography in total hip arthroplasty). Please clarify this in the Title, the Introduction and Aims as well as in the Discussion.

Methods

There is no definition of femoral neck anteversion (FNA). Several definitions have been published with highly variable results, depending on which anatomical reference points have been chosen, particularly the point of confluence for the femoral shaft and neck. Please elaborate on this and provide references.

Results

It is not completely clear which measurements are translated into an angle for FNA. You state that the mean asymmetry is about 1 mm which presumably is the mean of the mean 891 730 points of comparison in each patient? So – which of these points were used to create the FNA angle?

Discussion

Comparatively short, with only three new references introduced. This could be expanded somewhat. It is interesting to see that most of femoral torsion occurs in the neck.

References

Most cited references (for the Introduction) are concerned with 3D printing, which although interesting, is not the only field concerned with femoral symmetry. Other subjects could be introduced, such as the rationale for contralateral templating, the use of 3D templating and need for more exact measurements, other important landmarks for evaluation of symmetry.

Author Response

Dear reviewer

First of all, thank you so much for your feedback. With your excellent comments, we will improve it and will be easier to understand for the readers.

1-. In the study we have analyzed the femoral symmetry globally aligning it in the axial plane and then we have hypothesized the effect of the error in the femoral rotation, but the work is based on the analysis of a 3D volumetric asymmetry between two femurs globally. The problem with this kind of analysis using the Haussdorf-Besicovitch method is that the results are expressed in mm, so to better understand what does it mean, we make a correlation of how these mm could be expressed as degrees of femoral anteversion (147-154), but our goal is not to asses which is the intrapersonal variability of femoral anteversion but to asses the intrapersonal 3D global symmetry to analyze the origin of femoral anteversion, in line with another work that is under review right now for this journal (232-233). We have included a new figure to emphasize it and to show the differences between the two methods. This manual alignment method allows us to study the origin of the deformity and consequently, to establish where the osteotomy must be performed, for example, as we have explained in the new figure. It would be a very great job to analyze femoral and acetabular offset using 3D volumetric tools, and we will work on it, thank you so much.

  1. We completely agree with you. It’s difficult to establish which is the real femoral anteversion. We have clarified which femoral anteversion method we have used (Murphy’s method) and we have referenced it (77 and 148). Due to the difficult of the discussion and the main objective of the study we wouldn’t like to specifically discuss this point in the work, because we could distract the readers from the main purpose.

3-. To develop the translation from mm to degrees, we used the Murphy’s definition of femoral anteversion. Thanks for the suggestion, we have referenced it in the text (Ref 12).

4-. We have included some references about the use of 3D templating and the use of contralateral side to pre-bend plates prior to surgery, because it’s one of the most important application of the conclusions done by this study. With our results, we know that if we pre-bend a plate according with the mirror image of the contralateral side, the mean error respect its original shape will be 1 mm aprox. To understand this it’s important to take into account that we are not talking about an specific degree or an specific measurement, but its 3D global symmetry, what is a new concept of quality of global shape restoration, specially to locate where the osteotomy must be performed.

Finally, I would like to thank you your time and effort helping us to improve our work. This work is a bit different in respect with the traditional point of view of orthopedic surgery measurements, due to its 3D comprehension of the bones. We understand that I can be difficult to understand but thanks to the new software and technology, the 3D analysis will become more popular in the next few years, and with our results, we can stablish which is the error that we commit when taking the mirror image of the healthy contralateral bone as reference, which is one of the most used references in orthopedic surgery and traumatology. With your help our work has improved significantly, thank you so much.

Joan Ferràs-Tarragó

Vicente Sanchis-Alfonso

Round 2

Reviewer 2 Report

Thanks for addressing my comments.

I still disagree with point 1. The subtraction of an average manual error for the alignment of the same femur is not equivalent to perform the measurement without any manual error as claimed. First, placing a landmark in the same femur will have less inter and intraobserver variability that doing that for two different femurs with some level of asymmetry. So I would be less assertive regarding the claim that the asymmetry measurement is free of the manual alignment error, as this is just an approximate correction based on an average measurement for "same femur" landmarking.

Another point regarding the manual alignment is that in Figure 1 four landmarks are displayed and 4 points cannot be registered without any error (except for the single case where the four points are in the same plane). However, 3 points are used in the video demonstration and in this case the alignment of the points can be done perfectly and the only source of error would be the landmarking error. Please clarify this point and if necessary correct Figure 1.

Another comment:

New Figure 6. Please include units in histograms C and E. Please rephrase “The differences between both femurs are represented in a color code that is referenced in the lateral bar: the darker is the blue or the red color, the higher is the difference between both femurs” in the caption, the highlighted phrase is confusing.

Author Response

Dear Reviewer,

Thanks a lot for your interest in this work and for your effort helping us to improve it.

I completely understand what you meant with this point and I completely agree with you. We have defined the intrapersonal asymmetry after subtracting the mean of the manual alignment, but the intrapersonal asymmetry is under a confidence interval that could modify the value of the intrapersonal asymmetry. We have included this statement in the limitation and we mention that the intrapersonal asymmetry could be modified according with the manual alignment, being the intrapersonal variability predictable with the 95% IC of the results and more studies are needed to specifically assess the interobserver agreement (281-285), thank you so much for your review.

To select which points will be used as landmarks for the manual alignment tool we used the table-top method using a gravity simulation of the 3D program. This program simulates the effect of the gravity over a 3D object, in this case, over the femur. The effect is like if we put the femur over a table and we selected the contact points of the femur with the table. This makes easier to select the points. Depending of the femur morphology, the contact points of the femur with the table could vary, but the horizontal plane mustn’t. We didn’t add this explanation firstly because we thought that it was very technically, but It’s true that it will help to reproduce the experiment in case it was necessary. We have included a brief explanation in the figure 1 about it, just explaining that we used the concept of table top and we have referenced the article that described it to better understand it, thank you so much for your recommendation (104-106).

We have added the units in the histograms without specific values because the idea of these graphs is to show how the distribution can vary depending on the method used (not trying to show which is better or worse, as we discuss in the discussion) but to qualitatively show that the method can produce changes in the measurement. Finally, we have written better the phrase that was confuse (245-247).

Again, thank you so much for your time and effort helping us to improve our work.

Joan Ferràs-Tarragó

Vicente Sanchis-Alfonso
